# Unconventional oil and gas development and ambient particle radioactivity

Longxiang Li [1], Annelise J. Blomberg [1], John D. Spengler[1], Brent A. Coull[2], Joel D. Schwartz [1] & Petros Koutrakis[1✉]

Unconventional oil and natural gas development (UOGD) expanded extensively in the United States from the early 2000s. However, the influence of UOGD on the radioactivity of ambient particulate is not well understood. We collected the ambient particle radioactivity (PR) measurements of RadNet, a nationwide environmental radiation monitoring network. We obtained the information of over 1.5 million wells from the Enverus database. We investigated the association between the upwind UOGD well count and the downwind gross-beta radiation with adjustment for environmental factors governing the natural emission and transport of radioactivity. Our statistical analysis found that an additional 100 upwind UOGD wells within 20 km is associated with an increase of 0.024 mBq/m$^3$ (95% confidence interval [CI], 0.020, 0.028 mBq/m$^3$) in the gross-beta particle radiation downwind. Based on the published health analysis of PR, the widespread UOGD could induce adverse health effects to residents living close to UOGD by elevating PR.

[1] Department of Environmental Health, Harvard T.H Chan School of Public Health, Boston, MA 02114, USA. [2] Department of Biostatistics, Harvard T.H Chan School of Public Health, Boston, MA 02114, USA. ✉email: petros@hsph.harvard.edu

The extraction of crude oil and natural gas from the low-permeability unconventional geological accumulating formation (known as unconventional oil and natural gas development [UOGD]) expanded extensively over the past decade. As of 2017, over 120,000 onshore UOGD wells had been drilled via a practice involving directional drilling combined with multistage high-volume hydraulic fracturing (fracking)[1]. Meanwhile, numerous controversies have arisen, partially due to the potential harmful impacts on the local environment[2–7], and on the health of nearby residents[8–11].

Naturally occurring radioactive material (NORM) is a common by-product in Oil and Gas (O&G) production industry. The concentration of Uranium-238 in sedimentary formation rich in organic matter, such as black shale, is significantly higher than the background level of in earth's crust due to the natural attenuation process[12,13]. Before widespread UOGD, studies had detected above-background levels of Radium-226, a decay product of U-238, in the wastes of conventional oil and natural gas development (COGD)[14,15]. Regarding UOGD, enhanced levels of U-238 and Ra-226 have recently been detected in the produced water from unconventional hydrocarbon reservoirs[16,17], in the drill cuttings from the lateral drilling within the unconventional formation[18–20], in the impoundment sediments[21], in the soil of brine spill accident scene[22], and in the stream sediments near discharging sites[23]. Two studies in the Marcellus shale region found a positive association between UOGD activities and indoor levels of Radon-222, a gaseous decay product of Ra-226[24,25].

However, the influence of UOGD on the radioactivity of ambient particles (referred to as particle radioactivity [PR]) is not well understood. The particle-bound progeny of Radon-222 (referred to as radon in this study) contribute to the majority of PR[26,27]. Radon firstly decays into a chain of short-lived particle-reactive progeny. These short-lived radionuclides quickly react with the water molecules and atmospheric gases passing by, form ultrafine clusters and finally attach to airborne particles[28–30]. The short-lived progeny on the ambient particles then decay into two long-lived progeny, Lead-210 and Polonium-210, which respectively account for most of the beta- and alpha-radiation emitted by the particulate[26,31]. UOGD could influence local PR level by increasing the emission rate of radon. There is an increasing interest in the health effects of PR because the particle-bound Lead-210 and Polonium-210 tend to be deposited on the bronchial epithelium and expose neighboring cells to high-energy alpha particles that induce the carcinogenesis process[32,33]. Short-term exposure to PR has been associated with adverse health outcomes, including a decrease in lung function[34], an increase in blood pressure[35], and increased levels in biomarkers of inflammation[36,37].

In our study, we investigate the likely impact of UOGD on PR. Our statistical analysis demonstrates that upwind UOGD activities could significantly elevate the PR level in downwind communities. UOGD has a larger impact on PR, compared to COGD. The impact of UOGD on PR decreases gradually along with an increasing downwind distance. The results of our study contribute to the currently limited knowledge regarding the influence of UOGD on PR.

## Results

In this study, we analyzed 320,796 PR measurements carried out at 157 RadNet monitors across the continental United States from 2001 to 2017 (Fig. 1). We categorized these monitors into O&G RadNet monitors and other RadNet monitors, based on the existence of O&G extraction activities within 50 km. As summarized in Table 1, the national average PR level was 0.35 mBq/m³, with an interquartile range (IQR) from 0.22 mBq/m³ to

0.43 mBq/m³. O&G RadNet monitors had a higher average PR (0.39 mBq/m³; IQR: 0.26,0.47 mBq/m³), compared to the average PR of other O&G RadNet monitors (0.33 mBq/m³; IQR: 0.20,0.41 mBq/m³). Concerning PR emission-related environmental factors, O&G RadNet monitors had a higher ground surface U-238 level, and a higher percentage of air mass originated from the continent. For PR movement-dependent factors, O&G RadNet monitors had higher $PM_{2.5}$ concentrations, higher planetary boundary layer height (PBLH), higher wind velocity, and lower relative humidity.

After excluding wells without production records, there were 1,574,602 completed O&G wells by the end of 2017. Out of these O&G wells, 152,904 (9.7%) were UOGD wells, and 1,421,698 (91.3%) were COGD wells (Supplementary Fig. 1). Out of the UOGD wells, 4611 (3.0%) were within 20 km of RadNet monitors and 28,016 (18.3%) were within the 50 km buffer. UOGD expanded rapidly in all three subregions: Marcellus-Utica subregion, Permian-Haynesville subregion and Bakken-Niobrara subregion, during the study period (Supplementary Fig. 2, Supplementary Table 1). Fort Worth, Texas, had the highest average upwind UOGD count (mean 586, IQR: 504,661) within 20 km in 2017.

According to our linear mixed effect (LME) model, there was a statistically significant association between the downwind PR and the upwind UOGD activity. With adjustment for environmental factors regarding the natural emission and movement of PR, an additional 100 upwind UOGD wells within 20 km was associated with a 0.024 mBq/m³ increase in the level of PR (95% CI: 0.020, 0.028 mBq/m³) for a wind velocity of 1 m/s (Supplementary Table 2). Under the same wind condition, an additional 100 upwind COGD wells within 20 km was associated with a 0.004 mBq/m³ increase in PR (95% CI: 0.003, 0.004 mB1/m³). Regarding the magnitude of the impact, UOGD and COGD could elevate the PR level by up to 0.13 mBq/m³ and 0.029 mBq/m³, respectively. We also found significant negative interactions between wind velocity and the counts of both UOGD wells and COGD wells located upwind of RadNet monitors, suggesting lower influence when the wind is strong (Table 2).

Furthermore, we found that PR is significantly associated with upwind UOGD at every buffer distance of our study (Fig. 2). The influence of an additional 100 UOGD wells decreased gradually as the buffer radius increased from 20 to 50 km. An additional 100 UOGD wells within 50 km was associated with a 0.002 mBq/m³ increase in PR (95% CI: 0.002, 0.003 mBq/m³). Meanwhile, the association between the upwind COGD well count and PR was not statistically significant when the buffer radius is >20 km.

In the negative control analysis, we found PR level is also statistically associated with the number of UOGD wells within 20 km downwind of a RadNet monitor. However, the increase of PR associated with an additional 100 downwind UOGD wells within 20 km was 0.021 mBq/m³ (95% CI: 0.017, 0.024 mBq/m³), smaller than the impact of an additional 100 upwind UOGD well (0.024 mBq/m³, 95% CI: 0.020, 0.028 mBq/m³). We found that the per-well influence of upwind UOGD on downwind PR decreased gradually during our study period (Supplementary Note 2, Supplementary Table 3). After evaluating the sensitivity of our results, we argued that our results are not sensitive to the angle of the buffers (Supplementary Note 3, Supplementary Fig. 3). The upwind-downwind difference in the influence of UOGD wells is more significant when the angle of the circular sections is 60° compared to the quadrant buffer used in the primary analysis (Supplementary Table 4). We also found that our results are robust to omitting one of the RadNet monitors in the analysis (Supplementary Fig. 4).

We found that the PR level is statistically associated with the emission- and movement-dependent environmental covariates

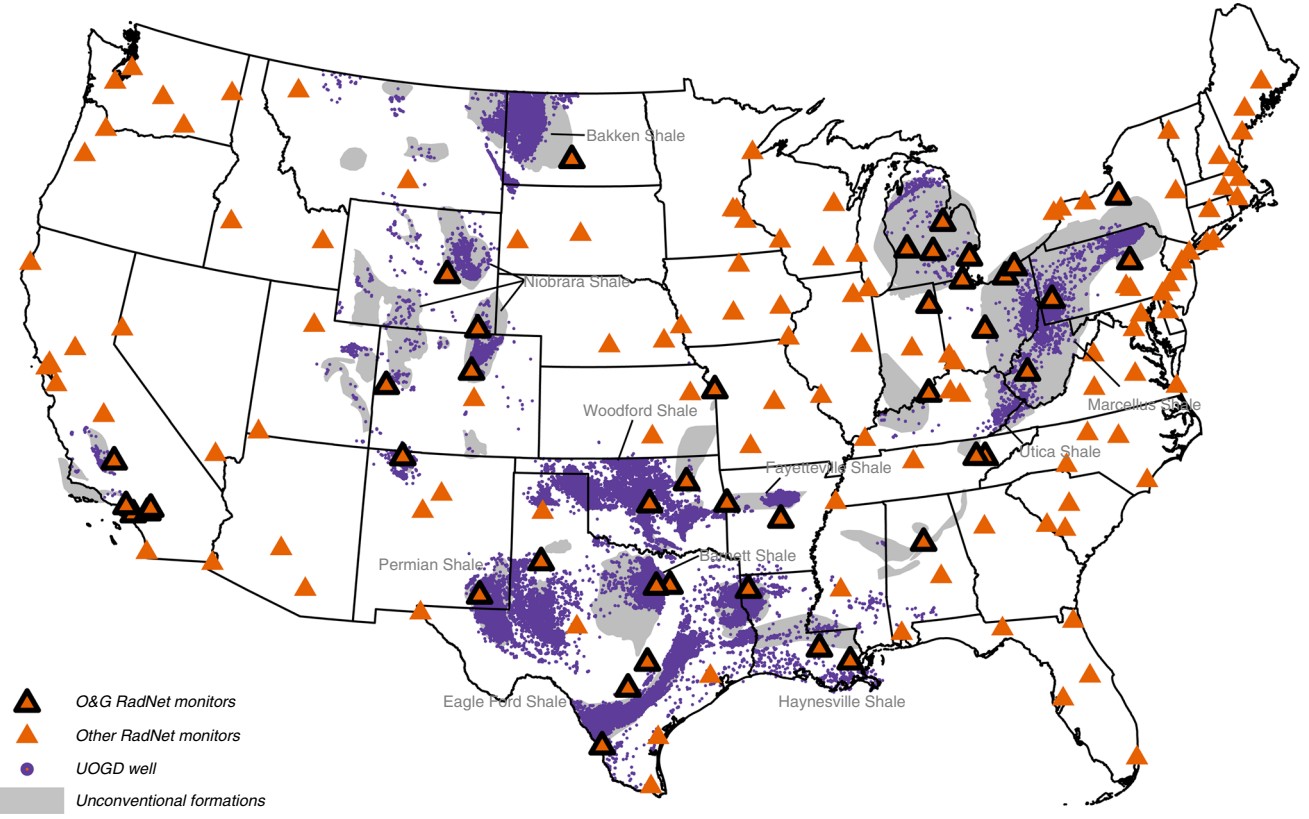

**Fig. 1 The location of RadNet monitors and the UOGD wells (completed by 2017) in the continental U.S.** The 157 RadNet monitors are categorized into O&G RadNet monitor (black edge) and Other RadNet monitors (no edge) based on whether a monitor is within 50 km of any O&G extractive activities.

**Table 1 Descriptive statistics of the ambient particle radioactivity and its environmental predictors.**

| Variable | Nationwide | O&G monitors | Other monitors |
|---|---|---|---|
| Monitors ($n$) | 157 | 43 | 114 |
| Observations ($n$) | 320,796 | 106,057 | 259,090 |
| PR (mBq/m$^3$) | 0.35 (0.22, 0.43) | 0.39 (0.26, 0.47) | 0.33 (0.20, 0.41) |
| $^{238}$U (ppm) | 1.82 (1.58, 2.17) | 1.91 (1.62, 2.19) | 1.74 (1.33, 2.12) |
| Origin of air mass (%) | 0.77 (0.57, 1.00) | 0.82 (0.75, 1.00) | 0.75 (0.56, 1.00) |
| PM$_{2.5}$ (μg/m$^3$) | 9.93 (5.74, 12.70) | 10.80 (6.49, 13.50) | 9.63(5.47, 12.05) |
| Soil moisture (ton/m$^2$) | 0.51 (0.42, 0.62) | 0.52 (0.42, 0.61) | 0.52 (0.43, 0.62) |
| Relative humidity (%) | 69.20 (62.1, 82.30) | 65.80 (55.4, 80.00) | 71.30 (64.7, 83.40) |
| Temperature (°C) | 13.90 (6.70, 22.10) | 13.90 (6.48, 22.32) | 13.48 (6.21, 21.63) |
| PBLH (km) | 0.91 (0.59, 1.13) | 0.98 (0.66, 1.18) | 0.89 (0.57, 1.11) |
| Wind speed (m/s) | 3.36 (2.07, 4.39) | 3.51 (2.14, 4.51) | 3.27 (1.96, 4.23) |
| Sunspots ($n$) | 66.76 (19.00, 97.30) | – | – |

Continuous environmental factors are summarized as the mean and the interquartile range (from 25th percentile to 75th percentile).
The unit of the gross-beta radiation is millibecquerel (mBq). 1 mBq/m$^3$ is equal to $1 \times 10^{-3}$Bq/m$^3$. 1 mBq/m$^3$ is also equal to 0.027 pCi/L.

(Table 2). There were significant positive correlations between PR and the ground surface concentration of U-238, the proportion of continent-sourced air mass, the number of sunspots, PM$_{2.5}$ concentration, and the inverse of PBLH. Meanwhile, PR is negatively associated with latitude, relative humidity, and soil moisture.

In our subregional analysis, we found significant heterogeneity among the three subregions regarding the influence of UOGD on PR (Table 3). In the Marcellus-Utica subregion, we found no evidence of a statistically significant association between upwind UOGD wells and downwind PR for any buffer distances investigated. Meanwhile, the association was significant for each buffer distance in the Permian-Haynesville subregion. In the Bakken-Niobrara subregion, the association was not statistically significant when the buffer radius is smaller than 30 km. However, when the buffer radius is 20 km, the impact of additional 100 UOGD wells in Marcellus-Utica subregion (0.180 mBq/m$^3$, 95% CI: −0.031 mBq/m$^3$, 0.390 mBq/m$^3$), though not significant, was greater than the impact in the other two subregions (Table 3).

## Discussion

In this study, we analyzed the radioactivity of airborne particles collected at 157 RadNet monitors across the continental United

**Table 2 The associations between PR and other environmental factors.**

| Term | Estimation | 95% CI | Details |
|---|---|---|---|
| U ($10^{-2}$) | 4.88 | (3.29, 6.48) | U-238 level in the ground surface material |
| origin ($10^{-2}$) | 7.13 | (6.85, 7.41) | The origin of the air mass. 1 indicates purely continental air mass; 0 indicates purely oceanic air mass |
| temp ($10^{-1}$) | −3.73 | (−4.48, −2.99) | The polynomial terms of air temperature. These are used to adjust for seasonality. |
| temp$^2$($10^{-3}$) | 1.20 | (0.93, 1.46) | |
| temp$^3$($10^{-6}$) | −1.27 | (−1.58, −0.96) | |
| year ($10^{-3}$) | −1.63 | (−2.21, −1.05) | The polynomial terms of the calendar year. These are used to control for long-term trend |
| year$^2$($10^{-6}$) | 9.29 | (−18.72, 37.31) | |
| PBLH$^{-1}$ | 16.53 | (15.50, 17.64) | Inverse of PBLH |
| pm ($10^{-2}$) | 1.12 | (1.11, 1.14) | Average concentration of $PM_{2.5}$ within 50 km from the RadNet monitor. |
| rhum ($10^{-4}$) | −3.04 | (−3.50, −2.58) | The relative humidity 2 meters above the surface |
| soilm ($10^{-5}$) | −4.84 | (−6.46, −3.21) | Liquid volumetric soil moisture in the top 1 m of soil |
| sun ($10^{-4}$) | 1.45 | (1.23, 1.67) | Monthly number of sunspots |
| lat ($10^{-3}$) | −6.84 | (−9.51, −4.18) | The latitude of the RadNet monitor. |
| vel ($10^{-3}$) | −0.91 | (−1.37, −0.45) | Wind velocity 10 m above the surface. |
| U•soilm ($10^{-5}$) | −6.57 | (−7.43, −5.72) | The interaction between U-238 concentration and soil moisture |
| sun•Lat ($10^{-5}$) | −2.52 | (−2.90, −2.15) | The interaction between the monthly count of sunspots and latitude |

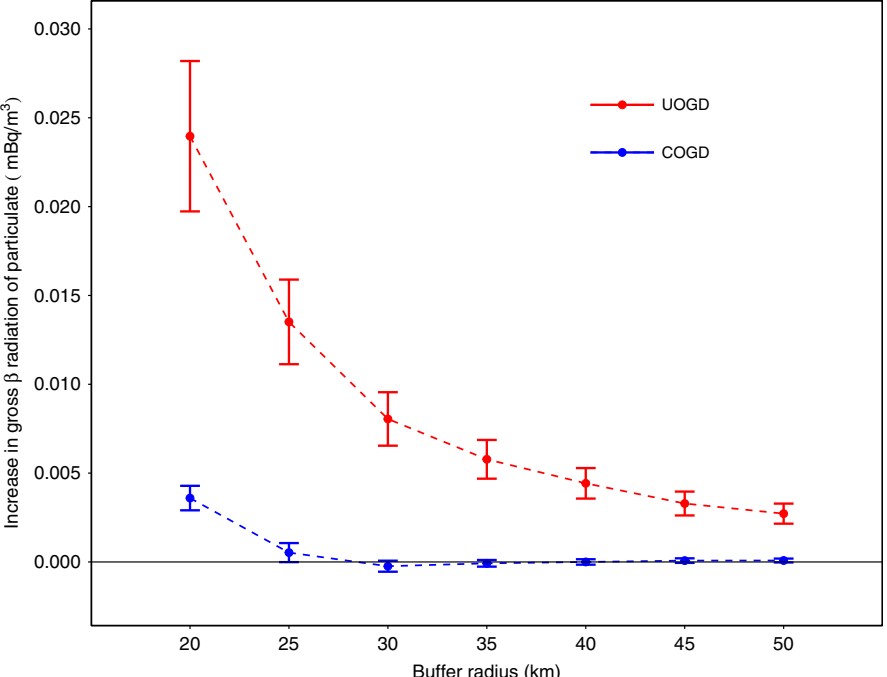

**Fig. 2 The association between upwind O&G production activities and downwind level of PR.** The increase in PR associated with an additional 100 UOGD wells (blue bars) and COGD wells (red bars) at multiple buffer distance. Effect estimations are visualized as the points and the 95% Cis are visualized as the bars. The source data of this figure is provided in Supplementary Table 2.

States from 2001 to 2017 (Fig. 1). To characterize upwind UOGD activities, we used the position and production records of 152,904 UOGD wells and counted the daily number of upwind UOGD wells. Our results added to the limited literature by evaluating the influence of UOGD on the radioactivity of ambient particles.

These associations suggested the existence of some pathways by which UOGD activities could release NORM into the atmospheric environment. Likely mechanisms include the fugitive release of natural gas, which contains a higher-than-background level of radon at wellheads, compressor stations, pipelines, and other associated facilities[38–40]; the management, storage, discharge and disposal of flow-back and produced water which is rich in NORMs[16,41–43]; the accidental spill or beneficial use of produced water in nearby communities[22]; the handling, transport, management, and disposal of radioactive drill cuttings[18,19].

The results of our negative control analysis suggested the potential transport mechanism of PR independent of atmospheric movement. This association could be explained by the increased off-site radon emission, or by the nearly isostropic dispersion of radon released on site under low-wind condition. To distinguish the contributions of these surface activities, more continuous measurements of PR, especially for some specific radionuclides, are needed at a finer spatiotemporal resolution.

Our results showed a remarkable distinction between the impacts of UOGD and COGD on PR. UOGD-specific processes, such as hydraulic fracturing and directional drilling, could potentially explain the larger associated impacts. The high-volume hydraulic fracturing process produced large volumes of flow-back water and drilling mud, which are subsequently stored in the temporary reserve pit adjacent to the drilling site.

**Table 3 The associations between PR and upwind UOGD well count in three subregions of our study extent.**

| Radius (Km) | Marcellus-Utica subregion | | Permian-Haynesville subregion | | Bakken-Niobrara subregion | | The whole study extent | |
|---|---|---|---|---|---|---|---|---|
| | Est ($10^{-2}$) | 95% CI ($10^{-2}$) | Est ($10^{-2}$) | 95% CI ($10^{-2}$) | Est ($10^{-2}$) | 95%CI ($10^{-2}$) | Est ($10^{-2}$) | 95% CI ($10^{-2}$) |
| 20 | 17.96 | (−3.06, 38.99) | 1.33 | (4.85, 7.80) | 1.26 | (−4.52, 13.79) | 2.40 | (1.97, 2.82) |
| 25 | 2.05 | (−2.95, 7.05) | 0.81 | (2.79, 4.12) | 0.93 | (−3.24, 11.07) | 1.35 | (1.11, 1.59) |
| 30 | 0.24 | (−1.16, 1.63) | 0.54 | (1.81, 2.87) | 3.61 | (3.87, 17.32) | 0.81 | (0.65, 0.96) |
| 35 | 0.11 | (−0.57, 0.79) | 0.38 | (1.20, 1.89) | 3.64 | (5.72, 12.93) | 0.58 | (0.47, 0.69) |
| 40 | 0.08 | (−0.35, 0.51) | 0.29 | (0.91, 1.42) | 2.12 | (3.42, 7.81) | 0.44 | (0.36, 0.53) |
| 45 | 0.07 | (−0.24, 0.38) | 0.21 | (0.65, 1.06) | 1.51 | (2.53, 5.80) | 0.33 | (0.26, 0.40) |
| 50 | 0.09 | (−0.17, 0.34) | 0.17 | (0.50, 0.86) | 1.05 | (1.81, 3.71) | 0.27 | (0.22, 0.33) |

The estimated influence (shown in columns named as Est) is presented as the increase in PR associated with an additional 100 UOGD wells within the radius.

Most UOGD production states allow the operator to close the reserve pit within up to one year after completing the drilling[44]. This practice potentially enables the NORMs in the produced water to decay into radon above the ground surface and release the radon into the ambient environment. The lateral drilling process produces large volumes of drill cuttings from the unconventional accumulating formation, whose levels of NORMs are higher than those produced during the vertical drilling stage. These drill cuttings are currently not considered hazardous wastes by U.S. EPA. The practice of beneficial use of drill cuttings and land treatment could potentially release radon into the ambient environment[45].

Our subregional analysis demonstrates remarkable heterogeneity in the estimated influences of UOGD in the three subregions. Due to a lack of monitors with UOGD wells nearby, the Marcellus-Utica subregional model did not have enough power to detect statistically significant associations. For a buffer radius of 20 km, the estimated influence of an additional 100 upwind UOGD wells is $17.96 \times 10^{-2}$ mBq/m$^3$ (95% CI: −3.06, $38.99 \times 10^{-2}$ mBq/m$^3$) in the Marcellus-Utica subregion, approximately seven times the estimated effects of a nationwide model. The difference is likely caused by the relatively few UOGD wells near the RadNet monitors in the Marcellus-Utica subregion (Supplementary Fig. 2). In the Bakken-Niobrara subregion, only two RadNet monitors (Casper, WY, and Navajo Lake, NM) have active UOGD wells around when the buffer radius is smaller than 30 km. When we enlarged the radius to 30 km, two additional RadNet monitors (Denver, CO, and Grand Junction, CO) had UOGD within the buffer, enabling us to identify the significant association (Supplementary Fig. 3).

Our results show a monotonic declining impact of O&G wells on PR as the buffer radius increases from 20 to 50 km (Fig. 2). The attenuation of radon after being emanated can explain this pattern. The trend indicates a more significant influence on the PR level of communities close to intensive UOGD activities. Limited by the accuracy of RadNet monitor location information, we did not estimate the impacts on a buffer distance smaller than 20 km. To tentatively extrapolate our results to these neighborhoods, we modeled the estimated influences as a power function of the radiuses with a negative exponent (Supplementary Note 5, Supplementary Fig 5). Based on this tentative extrapolation, an additional 100 UOGD wells within 10 km would be associated with an increase of 0.14 mBq/m$^3$. However, the result of this extrapolation should be interpreted cautiously. Monitors closer to UOGD wells are needed to validate this extrapolation.

One strength of our study is the nationwide monitor network of PR. The long-term measurement enables us to compare the current PR level with the baseline PR level in the absence of widespread UOGD. Furthermore, all filters were measured by NAREL using the same protocol, excluding the uncertainties induced by the heterogeneous devices operated by different labs. The other strength is the comprehensive database covering O&G activities. The Enverus database facilitated distinguishing the distinct impacts of UOGD and COGD on PR. In addition, we obtained diverse environmental covariates related to the natural emissions and transport of PR. Adjustment for these factors allowed us to draw conclusions explicitly related to the impacts of O&G development by explaining the natural variation of PR. The associations between PR and these environmental factors, as summarized in Table 2, are in agreement with the findings of previous studies[26,46].

One limitation of this study is that we only associated PR with the existence of completed O&G wells. Other construction-dependent factors may also influence the emission rate. However, the O&G wells with a detailed construction record are rare in our database, making it difficult to know the duration of construction, thus limiting our ability to investigate the construction-dependent association. Another limitation of our study is the simplification of the particle transport process. Our circular-sectional buffers are designed based on the Gaussian Dispersion Model. This calculation assumed a steady-state meteorological condition, which is reasonable in this case due to the short downwind distances. However, this computation could be improved by introducing advanced atmospheric dispersion models. Finally, we used drilling type as a proxy to whether an O&G well is UOGD or not. The method inevitably missed some vertical wells uncommonly completed by high-volume hydraulic fracturing.

Our results indicate the significant influence of UOGD on PR, a previously overlooked property of PM$_{2.5}$. Particulate-bound radon progeny continue releasing ionizing radiation after being inhaled and thus could induce systemic oxidative stress and inflammation, even at the levels observed in this study. Nyhan et al. (2018 and 2019) found that a 0.07 mBq/m$^3$ increase in 28-day average gross-beta radiation is associated with a 2.95 mm-Hg increase in diastolic blood pressure, a 3.94 mm-Hg increase in systolic blood pressure, a 2.41% decrease in forced vital capacity, and a 2.41% decrease in forced expiratory volume in Normative Ageing Study (NAS) population[34,35]. Blomberg et al.[37] reported that a 0.12 mBq/m$^3$ increase in seven-day average gross-beta radiation is associated with a 4.9% increase in C-reactive protein, a 2.8% increase in intercellular adhesion molecule-1, and a 4.3% increase in vascular cell adhesion molecule-1 in the same study popultaion. Jointly with these associations, our results suggest that an increase in PR due to the extensive UOGD may cause adverse health outcomes in nearby communities by elevating PR level (Supplementary Note 6). Further studies, especially those based on PR measurements close to UOGD activities, are needed to validate this exposure pathway.

Our analysis demonstrates that upwind UOGD activities could significantly elevate the PR level in downwind communities.

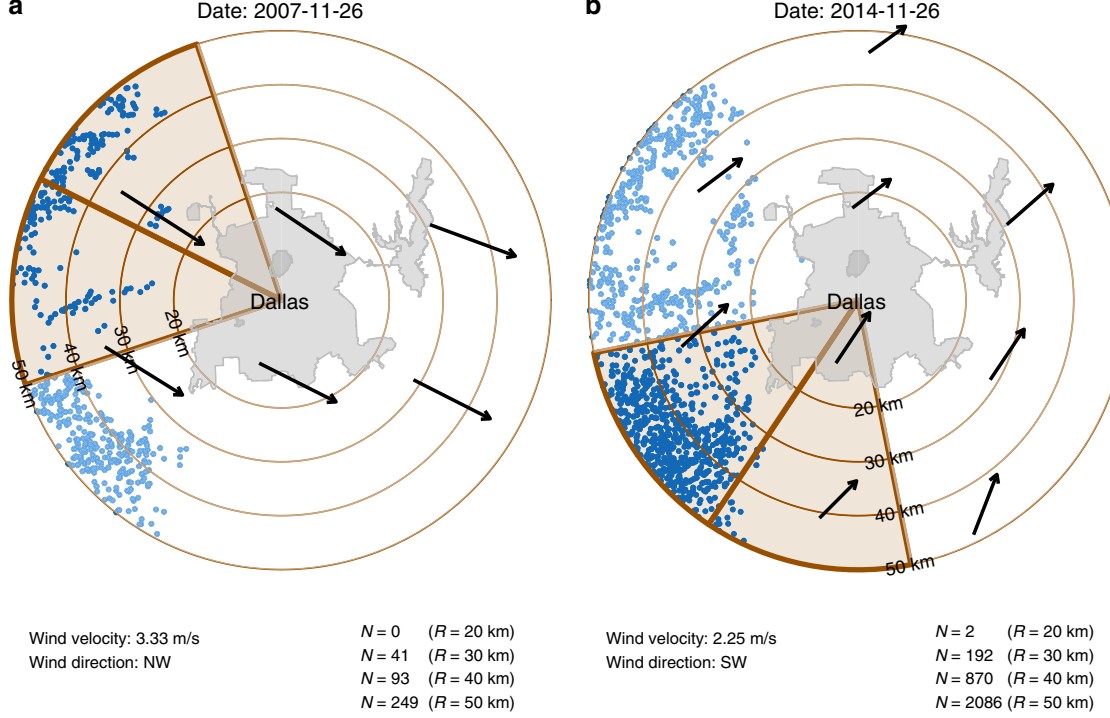

**a** Date: 2007-11-26

Wind velocity: 3.33 m/s
Wind direction: NW

| | |
|---|---|
| $N = 0$ | ($R = 20$ km) |
| $N = 41$ | ($R = 30$ km) |
| $N = 93$ | ($R = 40$ km) |
| $N = 249$ | ($R = 50$ km) |

**b** Date: 2014-11-26

Wind velocity: 2.25 m/s
Wind direction: SW

| | |
|---|---|
| $N = 2$ | ($R = 20$ km) |
| $N = 192$ | ($R = 30$ km) |
| $N = 870$ | ($R = 40$ km) |
| $N = 2086$ | ($R = 50$ km) |

**Fig. 3 Methods to calculate the number of UOGD wells positioned upwind of the RadNet monitor in two example days.** The example monitor is positioned at Dallas, TX. **a** The calculation of Nov-26-2007; **b** The calculation of Nov-26-2014.Due to the inaccessibility of the monitor's exact location, we used the geometric center of the city of Dallas, TX as a proxy. Based on daily wind direction (black arrows), we created the circular sectional buffer with a radius of 20 km and angle of 90 degrees. We created a series of buffers with radiuses ranging from 20 to 50 km, in order to investigate the scale dependency. We used the same method to count the daily number of COGD wells upwind of RadNet monitors.

UOGD has a larger impact on PR, compared to COGD. Based on previously published health effect analysis of PR, it is possible that the widespread of UOGD could induce adverse health effects to residents in proximity by elevating the PR.

## Methods

**Ambient PR measurements.** We obtained PR measurements carried out by the RadNet monitoring network, which is operated by the U.S. Environmental Protection Agency (EPA). This network measures the background environmental radiation levels in the air, precipitation, and drinking water under both routine and emergency conditions. During the study period from 2001 to 2017, 157 RadNet sites (Fig. 1) reported gross-beta measurements of various time lengths. Most RadNet monitors are located in metropolitan areas for better population coverage. At each site, total suspended particles (TSP) are collected using a high-volume sampler with a 4-inch diameter polyester fiber filter. Samplers are operated continuously for a 3- or 4-day integration. Filters are then sent to the National Analytical Radiation Environmental Laboratory (NAREL) for the measurement of gross-beta radiation[47,48]. To create quasi-daily values from the 3- or 4-day integrated samples, we assigned the same beta-radiation level to each day of the collection period.

**Unconventional oil and gas development data.** We obtained position and production information of O&G wells from Enverus (formerly Drillinginfo.com), a third-party data vendor used by the U.S. Energy Information Administration (EIA) to prepare monthly fossil fuel production and marketing reports. The comprehensive data coverage of Enverus is achieved by compiling the permits, construction logs, and production records from state agencies. Details about this data source were presented in a previous study[49]. Our dataset includes information for 2,159,858 wells hydraulically fractured from 01/01/1949 to 12/31/2017. We used drilling type information as the primary indicator of whether a well is targeting an unconventional accumulation formation or not[50]. Specifically, we considered horizontally drilled wells as UOGD wells and vertically drilled wells as COGD wells. Directionally drilled wells and wells without drilling type information were classified into UOGD or COGD wells based on their proximity to nearby UOGD wells and other secondary information using a Random Forest model (Supplementary Note 7).

We used the number of completed wells to characterize the intensity of O&G production activity. We identified a well as completed when the operator received the well from the driller. If this information is unavailable, we used the first

production date as a proxy. Considering the transport of airborne particles, we focused on the completed wells positioned upwind of the RadNet monitor. Specifically, we created a circular sectional buffer centered on the daily wind direction with an angle of 90° and a radius of 20 km (Fig. 3). We counted the numbers of UOGD and COGD wells, respectively, within the buffer on a daily basis to detect the different impacts of the two types of wells. We also created a series of circular sectional buffers at distances ranging from 25 to 50 km by 5 km intervals to evaluate the potential dependency of impact on the spatial scale. For security reasons, the exact location of RadNet monitors was not publicly accessible. We used the centroid of each RadNet city as a proxy to the sampling location (Fig. 3). Because of the potential spatial mismatch, we did not run any analysis on a spatial scale smaller than 20 km.

**Predictors of PR.** We collected data on environmental variables related to the emission of PR. To control for the emanation rate of radon from soil, we downloaded ground surface concentration of U-238 at a spatial resolution of 3 km from the United States Geological Survey[51]. PR is associated with the origin of air masses because the emanation rate of radon from the ocean is two orders lower than that from the continent[27]. To capture this pattern, we modeled four 72-hour back-trajectories (arrival time 06:00, 12:00, 18:00, and 24:00) of each RadNet monitoring sites using the Hybrid Single-Particle Lagrangian Integrated Trajectory (HYSPLIT) model[52]. The proportion of trajectory over the continent was used as a proxy to the origin of the air mass with 0% indicating a maritime air mass, while 100% indicating a continental air mass. Finally, we collected the monthly number of sunspots observed by the Royal Observatory of Belgium[53], as an indicator of the strength of solar activity. This information was used to adjust for the contribution of beta-emitting cosmogenic radionuclides originating from the upper atmosphere.

We also obtained environmental factors influencing the transport of PR. Due to the scavenger effect of aerosol on the short-lived progeny of radon, $PM_{2.5}$ concentration strongly influences the spatiotemporal distribution of radioactive isotopes in the atmosphere[27]. We downloaded daily $PM_{2.5}$ concentrations measured at EPA air quality monitors located within 50 km of RadNet sites and calculated the daily average if more than one measurement was carried out. The behavior of PR is driven by multiple meteorological factors, including wind velocity, relative humidity, PBLH, temperature, and soil moisture[26,27]. We obtained these variables from the North America Regional Reanalysis (NARR) dataset, with a spatial resolution of 32 km[54].

**Statistical analysis.** We applied LME models to investigate the association between PR and UOGD activities. Dependency between daily PR measurements

from the same monitor was controlled for by including monitor-specific random intercepts. We controlled for relevant environmental factors as fixed effects. We also adjusted for long-term temporal trend and seasonality by including polynomial terms based on the calendar year and temperature. We applied the LME as follows:

$$
\begin{aligned}
PR_{i,t} = {} & (c_0 + \gamma_i) + c_1 \cdot Num_{i,t} + c_2 \cdot U_i + c_3 \cdot origin_{i,t} + c_4 \cdot sun_t + c_5 \cdot pm_{i,t} + c_6 \cdot pblh_{i,t}^{-1} \\
& + c_7 \cdot rhum_{i,t} + c_8 \cdot soilm_{i,t} + c_9 \cdot vel_{i,t} + c_{10} \cdot lat_i + \sum_{p=11,12} c_p \cdot year_t^{p-10} \\
& + \sum_{p=13\cdots15} c_p \cdot temp_{i,t}^{p-12} + c_{16} \cdot sunspots_t \times lat_i + c_{17} \cdot soilm_{i,t} \times U \\
& + c_{18} \cdot Num_{i,t} \times vel_{i,t_i},
\end{aligned}
\tag{1}
$$

where $PR_{i,t}$ is PR level of site $i$ on day $t$; Num represents the number of upwind wells within the circular sectional buffer; $c_2$ to $c_4$ are the coefficients for the emission-dependent variables, including $U$ for the ground surface concentration of U-238, origin for the origin of the air mass and sun for the number of sunspots; $c_5$ to $c_9$ are the coefficients for transport-related environmental factors, including $pm$ for the concentration of PM$_{2.5}$, pblh$^{-1}$ for the inverse of HPBL, rhum for relative humidity, soilm for the moisture of soil and $vel$ for wind velocity; $c_{10}$ is the coefficient for latitude-dependent spatial trend; $c_{11}$ to $c_{15}$ are the coefficients for temporal trends terms, represented by the polynomial terms of the calendar year (year) and temperature (temp); $c_{16}$ to $c_{18}$ are the coefficients for interactions terms between environmental factors.

In the primary analysis, we associated daily PR levels with the daily number of upwind UOGD wells within 20 km using LME. To investigate the magnitude of UOGD's impact, we calculated the increase in PR associated with the 95% percentile of upwind UOGD well count. To investigate the dependency of effects on the transport distance, we counted the number of O&G wells within a series of circular sectional buffers at distances ranging from 25 to 50 km by 5 km intervals, and then estimated the effects for each buffer distances.

To investigate the influence of COGD wells, we associated daily PR with the upwind number of COGD wells within the same buffers. As a negative control, we counted the number of downwind UOGD wells within 20 km. We hypothesized that PR is associated with both upwind UOGD and downwind UOGD activities, but the influence should be smaller than that of upwind UOGD. Because of the pronounced changes in hydraulic fracturing practice in our study period[50], we explored the variation in the influence by restricting our model to four different sub-periods. To identify the potential regional heterogeneity in the effects of UOGD, we restricted our analysis to three separate subregions named after the shale formations underneath: Marcellus-Utica subregions, Permian-Haynesville subregion, and Bakken-Niobrara subregion (Supplementary Fig. 2). For sensitivity analysis, we first re-evaluated the associations by calculating the upwind UOGD wells within circular sectional buffers using another two central angles (60° and 120°). We then performed a leave-one-out sensitivity analysis to assess whether our estimates were sensitive to the omission of any single RadNet site.

We used the LME methods implemented in lme4 package (version 1.1–21)[55] in R (version 3.4.2)[56] to fit the models. The significance test was based on confidence intervals instead of p-values. The analysis was conducted on the Cannon cluster, supported by the Research Computing Group at Harvard University, Faculty of Arts and Sciences.

## Data availability

All source data for figures are available in the Supplementary Information. Explanatory data sources, including NARR meterological data (ftp://ftp.cdc.noaa.gov/Datasets/NARR/Dailies/monolevel/), ground surface uranium grid data (https://mrdata.usgs.gov/radiometric/), monitor-based PM2.5 measurements (https://aqs.epa.gov/aqsweb/documents/data_api.html), and number of sun black spot (http://www.sidc.be/silso/dayssnplot), are publicly accessible. The UOGD data from Enverus.com is subscription-only, thus cannot be shared publicly. If a token is provided, the raw data can be obtained from Enverus directly with the code we provided (55_Download_DI_Access.R). The aggregated data that support the findings of this study are available from the corresponding author upon reasonable request.

## Code availability

All model codes are available at: https://github.com/longxiang1025/Fracking_Radiation.

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

## Acknowledgements

This publication is made possible by U.S. EPA grant RD-835872. Its contents are solely the responsibility of the grantee and do not necessarily represent the official view of the U.S. EPA. Further, U.S. EPA does not endorse the purchase of any commercial products or services mentioned in the publication. We thank Drew Michanowicz and Jonathan Buonocore for their contribution to the work.

## Author contributions

P.K. initiated the study; L.L. synthesized data and performed research; L.L, B.A.C, and P.K. developed the model; and L.L., A.J.B., and P.K. wrote the manuscript. L.L., P.K., A.J.B., B.A.C., John D. Spengler, Joel D. Schwartz together interpreted the results.

## Competing interests

The authors declare no competing interests.
