## [Peer Review File · Nature Communications]

Reviewers' comments, first round:

Reviewer #1 (Remarks to the Author):

Summary

This manuscript uses EPA RadNet airborne particle radioactivity data to evaluate the association between conventional and unconventional completed wells and downwind particle radioactivity. They find support for the hypothesis that oil and gas extraction contribute to elevated airborne radioactivity, with stronger effects for unconventional wells. They include important confounding variables. I have several suggestions to improve the manuscript:

Major:

1. The introduction regarding radioactivity could be streamlined to more accurately reflect the pathway between UOG suggested by the authors. Uranium decays to radium, lead, and polonium, but that's not entirely clear as written.
2. For interpretability, I recommend reporting results in pCi/L (multiple results by 1/1000) rather than pCi/m³ because EPA recommendations are made in pCi/L. Alternatively, 37 Bq/m³.
3. It seems possible that PM_{2.5} is on the causal pathway between wells and particle gross beta activity; wells can produce PM_{2.5}. What happens if this variable is removed from the model?
4. Timing. It seems that timing of well exposures might matter quite a bit. For example, well pad construction and drilling would likely result in more upturn of soil and possibility of dispersion of radioactive materials than other phases, although the authors do provide hypotheses on other ways these materials could disperse (e.g., improperly stored drill cuttings or wastewater). The authors currently include any "completed" wells as the exposure of interest. Would it be possible to look at timing of completed spudding? In the intro, the authors mention having access to construction records.
5. Related to sub-regional analyses: Line 349: how do levels of uranium differ by the three regions? Any explanation for the larger coefficients at the 30 and 35 km radii in the Bakken? While the authors conclude that there's not much going on in the Marcellus, the point estimate is large (though not statistically significant) for the 20km radius. A bit more discussion would be helpful as this paper will certainly garner media attention. If distance to RadNet sites explains the lack of an association, what was the average distance between wells and RadNet sites in the Marcellus compared to the other formations?
6. The models contain a large number of covariates, is it possible to include a conceptual framework showing how the authors believe all these variables fit together? This could be added to the appendix.

Minor:

1. The paper requires editing for clarity, there are several typographical and sentence structure errors.
2. The last sentence of the abstract is strongly worded and implies causality. I am wavering over whether it's too strong.
3. Lines 29-32: this statement is misleading and only refers to oil and gas produced from tight formations, not all oil and gas as stated. Please update.
4. Line 33: typo: increased
5. Lines 40-42: please re-write sentence for clarity
6. Line 48: TENORM stands for Technologically Enhanced Naturally Occurring Radioactive Material
7. Figure 2: are the bars 95% CI? Please add to caption.
8. I see that relative humidity is included but wonder about rainfall. While perhaps highly correlated rainfall should modulate
9. Some minimal summary statistics would be helpful at the beginning of the results. How many wells on average were upwind of the monitoring sites at 20km buffer? How did this change over time? Did this change by season?
10. Lines 261-262: why select monitors with duration of 3+ years? Was this continuous or 3+ years between specific dates? Please add dates for which RadNet data was downloaded to this section. Also, please specify which years of data were included in regression models. This is not clear from equation 1.

11. Line 295: Were different circular sector degrees tested? For example, 45 degrees? You would anticipate a stronger response.
12. To clarify, a well would be included as upwind if, on day t, it was completed (this date is indicated in drilling info? Does complete mean it has been fracked?) and the wind blew the necessary direction? Providing an explicit definition would be helpful. Also defining "completed" and how this was determined for each well.
13. Thank you for providing the leave-one-out analysis, it's helpful.
14. Figure 1: could the formations in the sub-analysis be added here in grey? You may want to consider using another color scheme throughout because many readers are red/green colorblind.
15. As a negative exposure control you could test the effect of wells located downwind of the monitors, these should contribute much less or no exposure.

Reviewer #2 (Remarks to the Author):

Dear Authors,

Thank you for the opportunity to review your manuscript. This is a study that works to answer the question of the scale of impact of particulate radiation exposure through the air pathway sources from oil and gas development. This manuscript is novel in that no other study has attempted to do this to date and opens up another potential pathway of exposure to chemical constituents from O&G development that has not previously been quantified and thoroughly explored.

In general the manuscript is in good condition and the approach and findings are clear and well argued. As you will see in my annotation of your manuscript, I have asked a number of questions to clarify language around framing, definitions and results.

One contextualizing piece of information that could be discussed in this manuscript is whether the atmospheric enhancements of PR observed exceed known human health thresholds and appear to pose health risks to populations within the buffer distances investigated.

I am also curious to know if you might be able to use the distance decay that you observed across geographic space to be able to make educated predictions as to what the concentrations of PR would be at shorter distances from O&G operations. Your statement that they are likely higher seems reasonable, but as a reader, I am wondering if you can take this in a more quantitative direction.

Reviewer #4 (Remarks to the Author):

I have tried to read the article completely, but there is a great disorder in the presentation of the data and results. Its understanding is impossible, with many conceptual errors, being the information not well organized. The conclusions of the MS have not been demonstrated. Therefore, my decision has been to reject this MS in the current form.

Reviewer #5 (Remarks to the Author):

This is a well-written manuscript that attempts to address an interesting question, namely the extent to which unconventional oil and gas exploration contributes to the radioactivity of airborne particulate material. In fact, the manuscript goes a step further than that, linking findings of increased radioactivity to health impacts. Although the manuscript is interesting, there are some fundamental issues with the findings as presented and the extrapolations being made.

The following two statements from the abstract provide a useful basis for illustrating these issues.

Statement 1. "By analyzing airborne particle radioactivity records from the U.S EPA Radiation monitoring Network, we find that upwind unconventional oil and gas wells could elevate downwind

ambient particle radioactivity by up to 3.78×10^{-3} pCi/m³, or 44.8% over background level.”

This may be true, but is it of concern? The USEPA reports the average outdoor radon activity concentration in air to be 0.4 pCi/l (400 pCi/m³), with the average indoor radon activity concentration being 1.3pCi/l (1300 Bq/m³). The suggested potential maximum elevation in radioactivity of up to 0.00378 pCi/l (3.78 pCi/m³) due to unconventional oil and gas production is negligible when set in this context.

Whilst the Curie (Ci) is still used as a unit of measurement in North America, the Becquerel (Bq) is the SI unit most widely adopted for quantifying radioactivity and allows for direct comparison with other international datasets. Converting the maximum elevation in air activity concentration reported in the abstract to SI units gives 1.40×10^{-4} Bq/m³ (0.00014 Bq/m³). The World Health Organisation reports the global average outdoor radon to be in the range of 5 – 15 Bq/m³. Even recognising that the manuscript under review only reports gross beta activity (i.e. alpha and gamma activity are not considered), whether comparing with USEPA data or global data, the maximum elevation in activity reported in the present paper is negligible.

Statement 2. “These results confirm a link between unconventional oil and gas development and a new harmful environmental exposure.”

Here is perhaps the greatest area of concern within the paper. The authors are claiming that, what we now recognise to be a negligible potential elevation in the radioactivity of airborne particulates is leading to a new harmful exposure of the public. This is a bold claim and the data presented do not support this.

What matters when relating radioactivity to a biological endpoint is dose (radiation energy absorbed within biological tissue). The authors may argue that the difference is whether or not the activity is particle bound, but the reported negligible potential elevation in radioactivity will make little difference to the total dose to an individual. Even if we were to (i) approximate the slight elevations in alpha and beta emissions that may also have been identified if the authors had considered these, and (ii) apply radiation weighting factors to account for the relative biological effectiveness of these different emitters to enable the potential increase in equivalent dose to be calculated, the change in dose would still be negligible.

Why then are the authors proposing significant harm from a negligible elevation in dose? Looking at the references used in the paper to support the statements around the health impacts of particulate bound radioactivity, the three studies cited were all conducted by the authors of the paper under review here. It is good that the authors have opted to publish all three studies as Open Access, facilitating the evaluation of the scientific basis on which the claimed health impacts are based. However, when looking at these papers, it appears that two out of the three are focussing on two different potential effects within the same cohort of individuals and the papers discuss ‘associations’ rather than robust causal links. To lend weight to any claims that the authors may wish to make about the potential health impacts of elevated radioactivity of airborne particulate material, it would be helpful if the authors cited studies from other researchers that demonstrate causal links (if such studies exist).

Overall, as presented, I do not think that this paper is suitable for publication.

Response to Reviewers

Dear Reviewers,

Thank you for your insights and comments regarding our manuscript entitled “Unconventional Oil and Gas Development and Ambient Particle Radioactivity.” These comments, both major or minor, have helped us improve the manuscript significantly. We appreciated the opportunity to revise our manuscript according to your considered and careful reviews and are glad to present our revision.

We structure the response letter in two parts. We first reply to the comments offered by more than one reviewer. We then reply to the remaining comments of each reviewer individually.

Common comments

1. More than one reviewer suggested us to convert the unit of study from pCi/m³ to mBq/m³ for better interpretability.

Reviewer #1 wrote: For interpretability, I recommend reporting results in pCi/L (multiple results by 1/1000) rather than pCi/m³ because EPA recommendations are made in pCi/L. Alternatively, 37 Bq/m³.

Reviewer #5 wrote: While the Curie (Ci) is still used as a unit of measurement in North America, the Becquerel (Bq) is the SI unit most widely adopted for quantifying radioactivity and allows for direct comparison with other international datasets.

Reply: We thought this comment very helpful and converted the unit of all of our results. After conversion, the national average level of PR during our study period is 0.35 mBq/m³. Meanwhile, our primary result is that an additional 100 upwind UOGD wells within 20 km associates with an increment of 0.024 mBq/m³ in the gross-beta radiation downwind from the wells. For areas with 580 upwind UOGD wells within 20 km (the 95% percentile), UOGD could elevate the radiation level by 0.14 mBq/m³.

2. More than one reviewer pointed out that the UOGD-related enhancements in PR, even though they are measurable, may be too trivial to cause any health or environmental effects.

Reviewer #2 wrote: One contextualizing piece of information that could be discussed in this manuscript is whether the atmospheric enhancements of PR observed exceed known human health thresholds and appear to pose health risks to populations within the buffer distances investigated.

Reviewer #5 wrote: This may be true, but is it of concern? The USEPA reports the average outdoor radon activity concentration in air to be 0.4 pCi/l (400 pCi/m³), with the average indoor radon activity concentration being 1.3pCi/l (1300 Bq/m³). The suggested potential maximum elevation in radioactivity of up to 0.00378 pCi/l (3.78 pCi/m³) due to unconventional oil and gas production is negligible when set in this context.

The World Health Organisation reports the global average outdoor radon to be in the range of 5 – 15 Bq/m³. Even recognising that the manuscript under review only reports gross beta activity (i.e. alpha and gamma activity are not considered), whether comparing with USEPA data or global data, the maximum elevation in activity reported in the present paper is negligible.

Reply: We thought these comments very helpful and revised our manuscript accordingly.

First, we added a paragraph in the discussion section (Lines 288-300) associating our results with the previously published health effect analysis of particle radioactivity. The two publications found that an increase of 0.07 mBq/m³ increase in 28-day average associated with adverse health outcomes. Our study finds that UOGD could enhance the gross-beta level by up to 0.14 mBq/m³. The magnitude of the increase related to UOGD is above the threshold identified in these papers. As a result, we do not think this influence is negligible.

Second, in response to this comment, we re-wrote the first half of the paragraph (Lines 46-52) to clarify the relationship and difference between the activity concentration of radon and gross-beta radiation. The activity concentration of Pb-210 ($t_{1/2}=22.3$ years) is much lower than the concentration of Rn-222 ($t_{1/2}=3.8$ days) due to its much longer half-

life than Rn-222. As a result, the gross-beta radiation is also much lower than the activity concentration of radon. In the atmospheric environment, the activity concentration of radon (15 Bq/m³) is over forty thousand times higher than the gross-beta level (3.5×10^{-4} Bq/m³). A measurable increase in gross-beta suggests a measurable increase in radon level. Actually, enhanced levels of radon have been reported in Reference 24 and Reference 25.

Finally, we included a paper entitled “*The Role of Ambient Particle Radioactivity in Inflammation and Endothelial Function in an Elderly Cohort*” (Reference 37) which was written by a co-author of this paper and is recently accepted for publication by *Epidemiology*, a well-known journal in environmental epidemiology. This paper reports an elevated level of an inflammatory biomarker associated with a 0.14 mBq/m³ increase in the 7-day average gross-beta radiation. The results of this paper supports observable health effects of UOGD at environmentally relevant levels consistent with the present study.

Response to Specific Reviewers

For Reviewer #1

Dear Reviewer #1,

We are grateful for your considered and constructive comments about our manuscript. They have been incredibly helpful.

We are more confident about our results after conducting the extra sensitivity analysis you suggested. Our visualizations are more informative and reader-friendly after being revised according to your comments. We have extended our discussion to address the issues you indicated in our discussion sections.

We hope our revisions address your concerns.

Regards,

Longxiang Li

Comment 1: The introduction regarding radioactivity could be streamlined to more accurately reflect the pathway between UOG suggested by the authors. Uranium decays to radium, lead, and polonium, but that's not entirely clear as written.

Reply: We appreciate the reviewer's suggestion to clarify the potential mechanism by which UOGD could elevate PR. We re-wrote the paragraph accordingly (Lines 46-53).

Comment 2: For interpretability, I recommend reporting results in pCi/L (multiple results by 1/1000) rather than pCi/m³ because EPA recommendations are made in pCi/L. Alternatively, 37 bq/m³.

Reply: We addressed this comment in our reply to the shared comments, above.

Comment 3: It seems possible that PM_{2.5} is on the causal pathway between wells and particle gross beta activity; wells can produce PM_{2.5}. What happens if this variable is removed from the model?

Reply: In response to this comment, we repeated the analysis with slightly modification of the regression formula, removing the PM_{2.5} term in the equation. According to the simplified model, an additional 100 upwind UOGD wells are associated with a 0.025 mBq/m³ increase in PR (95% CI: 0.021, 0.03 mBq/m³). This is slightly higher than the estimated effect in the original analysis (0.024 mBq/m³, 95% CI: 0.020, 0.028 mBq/m³). We agree with you that PM_{2.5} is on the causal pathway. However, we determined not to add it to the main text because it does not influence our results remarkably.

Comment 4: Timing. It seems that timing of well exposures might matter quite a bit. For example, well pad construction and drilling would likely result in more upturn of soil and possibility of dispersion of radioactive materials than other phases, although the authors do provide hypotheses on other ways these materials could disperse (e.g., improperly stored drill cuttings or wastewater). The authors currently include any "completed" wells as the exposure of interest. Would it be possible to look at timing of completed spudding? In the intro, the authors mention having access to construction records.

Reply: Unfortunately, we cannot investigate the construction-dependent association because of the lack of detailed construction records. In response to your comment, we added this as a limitation in the discussion section (Lines 281-285).

Comment 5: Related to sub-regional analyses: Line 349: how do levels of uranium differ by the three regions? Any explanation for the larger coefficients at the 30 and 35 km radii in the Bakken? While the authors conclude that there's not much going on in the Marcellus, the point estimate is large (though not statistically significant) for the 20km radius. A bit more discussion would be helpful as this paper will certainly garner media attention. If distance to RadNet sites explains the lack of an association, what was the average distance between wells and RadNet sites in the Marcellus compared to the other formations?

Reply: We appreciate your comment regarding our discussion of the subregional analysis. In response, we extended the corresponding discussion paragraph (Lines 246-258). The average ground surface U-238 levels in three subregions are 2.00 ppm for Marcellus-Utica subregion; 1.71 ppm for Permian-Hayneville subregion; 2.30 for Bakken-Niobrara subregion. The reason for the large estimated coefficients in the subregional model is that there are not as many UOGD wells within 20 km in Marcellus-Utica region and Bakken-Niobrara region as in Permian-Hayneville region (Supplementary Information Table S1). It is reasonable to have a large estimated coefficient when the range of a predictor is small while the range of the dependent variable is kept unchanged. This heterogeneity highlights the importance of conducting the leave-one-out sensitivity analysis.

Comment 6: The models contain a large number of covariates, is it possible to include a conceptual framework showing how the authors believe all these variables fit together? This could be added to the appendix.

Reply: We have revised the related paragraphs in response to this suggestion. But we also added this into the main text because it significantly improves the clarity of our manuscript. We grouped all predictors in the model into three groups: O&G well count, radiation emission-related environmental factors (Lines 105-116), radiation transport-related factors (Lines 117-125), and spatial-temporal trends.

In addition, we added some comments on the formula of the regression model (Lines 132-141) and summarized these variables in Table 1 (Line 461). We presented the estimated associations between PR and these environmental factors in Table 2 (Line 468) and compared these coefficients with previous studies (Lines 278-280).

Comment 7: The paper requires editing for clarity, there are several typographical and sentence structure errors.

Reply: We have double-checked the revised manuscript and sent it to native English speaking colleagues for additional proof-reading.

Comment 8: The last sentence of the abstract is strongly worded and implies causality. I am wavering over whether it's too strong.

Reply: We agree. It is too strong. We re-wrote the abstract paragraph complying with the word count limit of Nature Communications (Lines 13-24). Most importantly, we replace that strong-word sentence with the conclusion sentence based on our findings and previously published results (Lines 22-24).

Comments 9: Lines 29-32: this statement is misleading and only refers to oil and gas produced from tight formations, not all oil and gas as stated. Please update.

Reply: We agree that this sentence overestimated the contribution of UOGD. We deleted this sentence in the revised version because our study focuses on the existence-dependent influence, instead of the production-dependent influence. (Lines 26-32).

Comment 10: Line 33: typo: increased

Reply: We corrected it, now in Line 30.

Comment 11: Lines 40-42: please re-write sentence for clarity

Reply: We re-wrote the sentence, in the revised version it is in Lines 50-56.

Comment 12: Line 48: TENORM stands for Technologically Enhanced Naturally Occurring Radioactive Material

Reply: We correct this in Lines 33-34 of the revision.

Comment 13: Figure 2: are the bars 95% CI? Please add to caption.

Reply: We added the note to all figures with bars representing the confidence interval (Figure 3, Lines 453-459).

Comment 14: I see that relative humidity is included but wonder about rainfall. While perhaps highly correlated rainfall should modulate.

Reply: We revisited the analysis in response to this comment. We understand that precipitation governs the wet deposition process and determines the residence time of aerosol. These effects may influence the particle-bound radioactivity level. However, the estimated coefficient for precipitation is not statistically significant, with adjustment for relative humidity, soil moisture, planetary boundary layer height, temperature, $PM_{2.5}$ concentration, and the origin of air mass. Likely, these environmental factors jointly explain the variation of precipitation.

Comment 15: Some minimal summary statistics would be helpful at the beginning of the results. How many wells on average were upwind of the monitoring sites at 20km buffer? How did this change over time? Did this change by season?

Reply: We have added paragraphs of summary statistics at the beginning of the results section in our revised manuscript (Lines 174-180). The temporal trend of upwind UOGD well count is summarized in Supplementary Information Section 1.3. Concerning seasonality, we calculated the monthly mean and standard deviation of upwind UOGD well count. The monthly trend is not significant, considering the large standard deviation. So, we did not include it in the manuscript.

Month	Average Upwind UOGD well count	Standard Deviation
1	70.3	165
2	76.5	167
3	68.8	156
4	70.8	161
5	71.3	165
6	71.5	176
7	78	179
8	84.6	182
9	73.2	163
10	68.7	165
11	66.1	157
12	71.5	167

Comment 16: Lines 261-262: why select monitors with duration of 3+ years? Was this continuous or 3+ years between specific dates? Please add dates for which RadNet data was downloaded to this section. Also, please specify which years of data were included in regression models. This is not clear from equation 1

Reply: Initially, we determined to use radiation measurements only from monitors with 3+ years of continuous monitoring records to eliminate the bias caused by the few monitors with short periods of operation. We agree that the 3 year cutoff seems arbitrary. Thus, in the revised manuscript, we included all radiation measurements without excluding the records from monitors with short periods of operation. We clarify the details in Lines 163-164. To test whether one monitor biased our result, we ran the leave-one-out sensitivity analysis (Supplementary Information Figure S5).

Comment 17: Line 295: Were different circular sector degrees tested? For example, 45 degrees? You would anticipate a stronger response.

Reply: In response to this comment, we added a sensitivity analysis to the statistical analysis section (Lines 153-155). Then the results are now mentioned in the discussion section (Lines

202-203) and detailed in the supplementary information Section 2.2, Figure S4. Our results are not sensitive to a change in the angle of the circular sector. In addition, the influence of UOGD is stronger when we shrink the angle from 90° to 60°, as expected.

Figure S1. The increment in PR associated with an increase of 100 upwind UOGD wells in circular sectional buffers with different central angles.

Comment 18: To clarify, a well would be included as upwind if, on day t, it was completed (this date is indicated in drilling info? Does complete mean it has been fracked?) and the wind blew the necessary direction? Providing an explicit definition would be helpful. Also defining “completed” and how this was determined for each well.

Reply: To clarify the definition of “completed” well, we add Lines 93-95. A well is completed when all construction jobs are finished (including fracking) and is ready to be transferred to the operator for production. We also included Figure 2 to demonstrate how we calculate the upwind completed UOGD well count.

Comment 19: Figure 1: could the formations in the sub-analysis be added here in grey? You may want to consider using another color scheme throughout because many readers are red/green colorblind.

Reply: We have revised the figure according to your comment. The revised Figure 1 is attached here for a quick review.

We also adjust all the figures to make them more color-blind friendly. Thank you for pointing this out.

Comment 20: As a negative exposure control you could test the effect of wells located downwind of the monitors, these should contribute much less or no exposure.

Reply: We appreciate the suggestion and conducted the analysis accordingly. It is briefly included in the statistical analysis section (Lines 149-150) and the results are presented in Lines 198-202. The downwind UOGD well count is also positively associated with PR. But the influence is smaller, as expected. The positive association is probably due to the close relationship between the upwind UOGD well count and the downwind UOGD well count (spearman $R^2=0.86$).

For Reviewer #2

Dear Reviewer #2,

Thank you so much for your constructive comments.

We feel that the manuscript is improved significantly by addressing your comments. We tentatively extrapolate our results to a smaller spatial scale. We also clarify the health implications section by comparing our results with the previously published health effect analysis.

We sincerely hope our revisions address your concerns regarding our previous manuscript.

Regards,

Longxiang Li

Comment 1: One contextualizing piece of information that could be discussed in this manuscript is whether the atmospheric enhancements of PR observed exceed known human health thresholds and appear to pose health risks to populations within the buffer distances investigated.

Reply: We have substantially revised our manuscript to address these concerns. After converting the unit of our results, an additional 100 upwind UOGD wells within 20 km associates with a 0.024 mBq/m³ increase in the level of PR (95% CI: 0.020, 0.028 mBq/m³). In areas with 580 upwind UOGD wells within 20km, the increase is approximately 0.14 mBq/m³. This enhancement is above the levels that are found associated with significant adverse health effects in previous studies (0.07 mBq/m³ in 28-day average and 0.14 mBq/m³ in 7-day average). We detailed these comparisons in Lines 290-302.

Comment 2: I am also curious to know if you might be able to use the distance decay that you observed across geographic space to be able to make educated predictions as to what the concentrations of PR would be at shorter distances from O&G operations. Your statement that they are likely higher seems reasonable, but as a reader, I am wondering if you can take this in a more quantitative direction.

Reply: In response to the comment, we made a tentative extrapolation of our results to a smaller spatial scale, detailed in Lines 264-269 of the revised manuscript and Supplementary Information section 3.1, Figure S6. We attach the figure here for a quick review. We used the power function of downwind distance with a negative exponent to model the distance-decay. The exponent with the best fit is -2.5. Based on this decay gradient, an additional 100 UOGD wells within 10 km is associated with an increase of 0.14 mBq/m³.

Figure S2. The observed distance-dependent decay of the estimated effects and the modeled distance decay by power functions with negative exponents.

Comment 3: Lines 25. The sentence “The results confirm a link between UOGD and a new harmful environmental exposure” is too strong

Reply: We agree. It is too strong. We re-wrote the abstract paragraph complying with the word count limit of Nature Communications (Lines 13-24). Most importantly, we replace that strong-word sentence with the conclusion sentence based on our findings and previously published results (Lines 22-24).

Comment 4: Lines 28 needs to be more specific.

Reply: We agree it needs clarification. We have revised it accordingly in Lines 13-14.

Comment 5: Lines 28-31 We wrote “By the end of 2018, directionally drilled wells completed with hydraulic fracturing (commonly referred to as unconventional oil and gas [UOG] wells) accounted for 96% and 97% of the domestic crude oil and natural gas production, respectively.”

Your comments: The term unconventional oil and gas development has two main definitions:
The Engineering Definition: The application of directional drilling and hydraulic fracturing
The Geological Definition: The development of hydrocarbons from low permeability reservoirs and primarily from source rock. What you are referring to is the Geological definition and EIA states that ~96% of O&G from TIGHT OIL AND SHALE GAS PLAYS is attributable to the application of horizontally drilled wells. This is different from the calculation of the proportion of hydrocarbon production attributable to horizontal drilling and hydraulic fracturing which is much lower (i.e., there is a lot of oil and gas that is still produced with the application of other techniques such as enhanced oil recovery, etc.

Reply: We agree that this sentence overestimated the contribution of UOGD. We deleted this sentence in the revised version because our study focuses on existence-dependent influence, instead of the production-dependent influence. (Lines 26-32).

Comment 6: Lines 33. Additional references are recommended.

Reply: We agree and additional references are included in Lines 30-32 of the revised manuscript.

Comment 7: Lines 39-41. We wrote, “If inhaled, these radioisotopes, such as Lead-210 (210Pb) and Polonium-210 Po, a known strong carcinogen), on the particulate contribute to internal radiation when they decay and emit α -, β -, and γ - radiations.”

Your comment: This sentence does not make sense.

Reply: We have revised this for clarity in Lines 50-56.

Comment 8: Lines 42-43. We wrote, “Several recent cohort studies have found associations between PR and short-term health outcomes, including a decrease in forced vital capacity⁸, an increase in systolic and diastolic blood pressure ...”

Your comment: Perhaps put this into more plain language such as “lung function”.

Reply: We have revised accordingly in Lines 56-58.

Comment 9: Line 52. Additional reference is recommended.

Reply: We agree; additional references are now included in Lines 229.

Comment 10: Line 57-67. We wrote “Compared with COG wells, UOG wells have a larger theoretical potential to elevate PR...”

Your comment: Your examples below are part of the issue, but also is the probable driver that concentrations of NORMs in deeper source rock are higher. Please mention this here with appropriate citations.

Reply: We have revised to clarify the difference between UOGD and COGD in Lines 234-245. We cited reference regarding the higher-than-background level of NORM in sedimentary rock rich in organic matter, such as black shale. We did not find a publication supporting the depth-dependent distribution of NORMs. However, it is probably true because deeper formations are older than the shallower formations. Please also refer to our reply to **Comment 11**, because these two comments are closely related.

Comment 11: Line 59. We wrote “UOG wells produces larger volumes of drill cuttings and flowback water compared to COG wells, because of the combination of lateral drilling and hydraulic fracturing.” Line 63. We wrote “Increasing lateral drilling length and ”

Your comment: These volumetric differences are much more tied to the length of the wells than to the direction of drilling a hydraulic fracturing

Reply: We agree that the volume of drill cutting is governed by the drilling length. In response to the comment, in the revised manuscript we write that lateral drilling creates a large volume of cuttings sourced from the highly radioactive formation, while the vertical drilling mostly goes through formations of lower radioactivity (Lines 241-243).

Comment 12: Line 63. We wrote “Increasing lateral drilling length and water usage could exacerbate the condition in the future”

Your comment: Why just LATERAL drilling length? Why not just say increased length of wellbores? Why would increased water usage alone increase risk of PR production?

Reply: We have revised the statement to clarify the mechanism in Lines 234-245. Briefly, water usage alone does not matter a lot. The large volume of produced water may be problematic.

Comment 13: Line 108. We wrote “However, these results are not directly comparable, as remarkably changing number of wells at different buffers, or differences in the number of COG and UOG wells, may influence the magnitude of the results”

Line 115. We wrote “This value is selected to represent the number of upwind O&G wells in regions with intensive oil and gas development and is also closely associated with the area of buffer,”

Your comment: This discussion is confusing

Reply: We have revised to clarify the discussion regarding the distance-decay influence (Lines 259-262).

Comment 14: Lines 132-138. We discussed the potential leakage mechanism of PR from UOGD.

Your comment: I would not consider all of these “off-site” as many of these operations occur on the well pad. Perhaps refer to this class of activities as “surface activities, transmission and waste handling”?

Reply: We sincerely appreciate your suggestions to improve this discussion. We revised this paragraph accordingly (Lines 225-233).

Comment 15: Lines 201. We cited a report published on EIA’s website with a wrong access date as “Accessed: 10th December 2019”

Your comment: This does not seem possible. Please note the actual date accessed.

Reply: We accessed the report on 12th Oct 2019. However, this reference was removed in our revised manuscript in response to Comment 5, above.

For Reviewer #5

Dear Reviewer #5,

Thank you for sharing with us your helpful and constructive critical comments. We agree that some wording in the previous manuscript may have been overly strong and have revised accordingly.

If we understand correctly, your primary concern is whether UOGD could elevate PR to a “harmful” level. The short answer should be yes, considering the results of three peer-reviewed and published studies. We agree that studies conducted by outside researchers may provide additional insight. However, this is a relatively new field with limited existing knowledge. The first paper was published less than three years ago. A lack of previous studies may reflect more on the novelty of this area than on its value or importance.

I hope our detailed reply to your comments could at least in part address your concerns. We look forward to your feedback.

Best regards,

Longxiang Li

Comment 1: While the Curie (Ci) is still used as a unit of measurement in North America, the Becquerel (Bq) is the SI unit most widely adopted for quantifying radioactivity and allows for direct comparison with other international datasets.

Reply: In response to this comment, we have converted all the results into the SI units. After conversion, the national average level of PR during our study period is 0.35 mBq/m³. Meanwhile, our primary result is that an additional 100 upwind UOGD wells within 20 km is associated with an increase of 0.024 mBq/m³ in the gross-beta radiation downwind from the wells. For areas with 580 upwind UOGD wells within 20 km (the 95% percentile), UOGD could elevate the radiation level by 0.14 mBq/m³, in agreement with the calculation by the reviewer.

Comment 2: This may be true, but is it of concern? The USEPA reports the average outdoor radon activity concentration in air to be 0.4 pCi/l (400 pCi/m³), with the average indoor radon activity concentration being 1.3pCi/l (1300 Bq/m³). The suggested potential maximum elevation in radioactivity of up to 0.00378 pCi/l (3.78 pCi/m³) due to unconventional oil and gas production is negligible when set in this context.

The World Health Organisation reports the global average outdoor radon to be in the range of 5 – 15 Bq/m³. Even recognising that the manuscript under review only reports gross beta activity (i.e. alpha and gamma activity are not considered), whether comparing with USEPA data or global data, the maximum elevation in activity reported in the present paper is negligible.

What matters when relating radioactivity to a biological endpoint is dose (radiation energy absorbed within biological tissue). The authors may argue that the difference is whether or not the activity is particle bound, but the reported negligible potential elevation in radioactivity will make little difference to the total dose to an individual. Even if we were to (i) approximate the slight elevations in alpha and beta emissions that may also have been identified if the authors had considered these, and (ii) apply radiation weighting factors to account for the relative biological effectiveness of these different emitters to enable the potential increase in equivalent dose to be calculated, the change in dose would still be negligible.

Reply: We appreciate this critical argument. .

We have revised the first half of the paragraph (Lines 46-52) to clarify the relationship and difference between the activity concentration of radon and the particle-bound gross-beta

radioactivity. The activity concentration of Pb-210 ($t_{1/2}=22.3$ years) is much lower than the concentration of Rn-222 ($t_{1/2}=3.8$ days) due to its much longer half-life than Rn-222. As a result, the gross-beta radiation is also much lower than the activity concentration of radon. In the atmospheric environment, the activity concentration of radon (3.7 Bq/m^3) is over ten thousand times higher than the gross-beta level ($3.5 \times 10^{-4} \text{ Bq/m}^3$). Due to the noble nature of radon, its *carcinogenic* health effects are primarily caused by its radioactive progeny instead of radon itself. Gross-beta radiation, mostly contributed by Pb-210, the first long-lived progeny of the decay products of radon and is theoretically closer to the health outcome than the activity concentration of radon. Even though the absolute increase in beta radiation associated with UOGD is small, it represents approximately a 45% increase over the background level, which may not be neglectable.

Particle-bound radioactivity is the radioactive character of the ambient. The behavior of $\text{PM}_{2.5}$ in our body plays an essential role in calculating the effective dose. However, our knowledge in this field is still limited. That is why we used population-based studies to evaluate the health effects of PR instead of a physical model. According to our previous published papers, a 0.14 mBq/m^3 increase in 7-day average PR and a 0.07 mBq/m^3 increase in 28-day average PR are both significantly associated with increased blood pressure, lower lung function and increase levels of inflammatory biomarker. The UOGD-related enhancement in PR is up to 0.14 mBq/m^3 , above these thresholds.

Comment 3: Why then are the authors proposing significant harm from a negligible elevation in dose? Looking at the references used in the paper to support the statements around the health impacts of particulate bound radioactivity, the three studies cited were all conducted by the authors of the paper under review here. It is good that the authors have opted to publish all three studies as Open Access, facilitating the evaluation of the scientific basis on which the claimed health impacts are based. However, when looking at these papers, it appears that two out of the three are focussing on two different potential effects within the same cohort of individuals and the papers discuss ‘associations’ rather than robust causal links. To lend weight to any claims that the authors may wish to make about the potential health impacts of elevated radioactivity of airborne particulate material, it would be helpful if the authors cited studies from other researchers that demonstrate causal links (if such studies exist).

Reply: We eagerly anticipate the growth of interest in and research into this area by others in our field. However, to our best knowledge, there are not yet any published studies conducted by other authors regarding the health effects of ambient PR. Our research group is the first to comprehensively investigate this potentially overlooked feature of PM_{2.5}, and we have published at least 6 papers on PR studies including at least 3 cohorts in peer-reviewed journals to date, several of which are open access. We believe that this limitation is (1) likely to be short-lived as other researchers publish papers in this area, and (2) at least partially balanced by the peer-review process and the constructive and critical comments of reviewers and editors.

Reviewer comments, second round

Reviewer #1 (Remarks to the Author):

The authors were very responsive to reviewer comments. Related to whether the observed magnitude of the change in PR associated with additional UOGD wells is relevant to health. The authors of this study have 10+ peer-reviewed articles finding associations between 0.05 to 0.10mBq/m³ increases in PR and adverse health outcomes (e.g., elevated blood pressure, decreased lung function). I was unable to locate any health articles on this topic not authored by the present study team. However, this does not mean their findings are not important and relevant to health. In summary, based on the current team's prior work on PR and health, the change in PR induced by UOGD appears relevant to human health. I have just a few additional comments below:

Major

1. As written, the results lead the reader to believe that there was not a relationship between UOGD and PR in the Marcellus, however, as discussed later, this is likely due to small sample size. Indeed, the point estimate within 20km is much larger in the Marcellus than elsewhere. Suggest conveying this information more clearly in the results.
2. Does the relationship between wells and PR change over time? I ask this because processes have changed dramatically in the Marcellus over time, for example. Early on, pits were unlined and often percolated (bubbled to induce evaporation of wastewater, yikes). These practices have slowed and unlined open pits are less common. I am less familiar with practices in the other formations but this might be of interest to readers.
3. Please add a bit more discussion regarding the negative control analysis. The main analyses seem to suggest that more wind reduces PR, so perhaps wind direction makes less of a difference at the lower speeds relevant to PR transport? The wind resolution data was only 32 km and the buffer was 20km, so this is also somewhat problematic. You use average daily wind direction, so it's also possible at low wind speeds that the wind blew both ways that way. Regardless, more discussion is needed as the present negative control without further explanation appears to negate the findings somewhat.

Minor

1. Abstract: Unconventional Oil and Natural Gas should be sentence case
2. Abstract: please provide confidence interval for 0.024 mBq/m³
3. Line 85: these are stimulated wells, so indicating hydraulically fractured? Your definition thus requires both horizontal/directional drilling and fracturing to be considered UOGD? This makes sense in many places, but, for example, in California, hydraulically fractured wells tend to be shallower and also vertical. So, this method would miss these wells.
4. Line 179: list the three subregions.
5. What is this sentence based on explicitly? Regarding the magnitude of the impact, UOGD and COGD could elevate the PR level by up to 0.13 mBq/m³ and 0.029 mBq/m³, respectively.

Reviewer #5 (Remarks to the Author):

Thank you for the opportunity to read the revised manuscript and rebuttal letter. It is pleasing to see that the authors found the comments from all 5 reviewers beneficial in revising the manuscript and they have made appropriate modifications. I am still not convinced by the suggested health impacts, but the authors have at least toned down their statements in relation to this. I think it appropriate that this manuscript is now published so that others can then read and judge for themselves.

Response to Reviewers

Dear Referees,

Thank you for your insights and comments regarding our revised manuscript entitled “Unconventional Oil and Gas Development and Ambient Particle Radioactivity.” We thought all your comments constructive. That is why we revised our manuscript accordingly immediately after receiving your feedbacks. We appreciated the opportunity to improve our manuscript and are glad to share with you our revision.

Best regards,

Longxiang Li

Dear Referee #1,

We are grateful for your considered and constructive comments about our manuscript. They have improved our manuscript remarkably.

According to your comments, we have extended our discussion to address the issues about the negative control analysis and subregional analysis. In addition, we thought the temporal analysis of the influence of UOGD important and conducted it.

We hope our revisions address your concerns.

Regards,

Longxiang Li

Reviewer #1 (Remarks to the Author):

The authors were very responsive to reviewer comments. Related to whether the observed magnitude of the change in PR associated with additional UOGD wells is relevant to health. The authors of this study have 10+ peer-reviewed articles finding associations between 0.05 to 0.10mBq/m³ increases in PR and adverse health outcomes (e.g., elevated blood pressure, decreased lung function). I was unable to locate any health articles on this topic not authored by the present study team. However, this does not mean their findings are not important and relevant to health. In summary, based on the current team's prior work on PR and health, the change in PR induced by UOGD appears relevant to human health. I have just a few additional comments below:

Major

1. As written, the results lead the reader to believe that there was not a relationship between UOGD and PR in the Marcellus, however, as discussed later, this is likely due to small sample size. Indeed, the point estimate within 20km is much larger in the Marcellus than elsewhere. Suggest conveying this information more clearly in the results.

Reply: We appreciate referee's suggestion to clarify the results of subregional analysis. We revised the paragraph according to referee's suggestion (Lines 261-273).

2. Does the relationship between wells and PR change over time? I ask this because processes have changed dramatically in the Marcellus over time, for example. Early on, pits were unlined and often percolated (bubbled to induce evaporation of wastewater, yikes). These practices have slowed and unlined open pits are less common. I am less familiar with practices in the other formations but this might be of interest to readers.

Reply: We appreciate referee's comment and added the analysis of temporal variation in the influence of UOGD in both Methods (Lines 153-155), Results (Lines 207-209) and Supplementary Information (Section 2.2).

3. Please add a bit more discussion regarding the negative control analysis. The main analyses seem to suggest that more wind reduces PR, so perhaps wind direction makes less of a difference at the lower speeds relevant to PR transport? The wind resolution data was only 32 km and the buffer was 20km, so this is also somewhat problematic. You use average daily wind direction, so it's also possible

at low wind speeds that the wind blew both ways that way. Regardless, more discussion is needed as the present negative control without further explanation appears to negate the findings somewhat.

Reply: We appreciate referee's comments that more discussion about the results of negative control analysis is necessary. We expanded our interpretation of the results in discussion section (Lines 242-245). In addition, we think it helpful to detail the negative control analysis in Methods section (Lines 150-153)

Minor

1. Abstract: Unconventional Oil and Natural Gas should be sentence case

Reply: We appreciate referee's comment and correct the mistake in both abstract and main text (Line 13)

2. Abstract: please provide confidence interval for 0.024 mBq/m³

Reply: We appreciate referee's suggestion and add the .95 confidence interval in abstract (Line 21~22)

3. Line 85: these are stimulated wells, so indicating hydraulically fractured? Your definition thus requires both horizontal/directional drilling and fracturing to be considered UOGD? This makes sense in many places, but, for example, in California, hydraulically fractured wells tend to be shallower and also vertical. So, this method would miss these wells.

Reply: We appreciate referee's comment. Yes, the stimulated wells mean hydraulically fractured wells. We replace "stimulated" with "hydraulically fractured" to get rid of confusion. We admit that our random forest model misses the vertical UOGD wells in CA. This is an important limitation which we did not discuss. We add some discussion in the last paragraph in the discussion section. (Lines 304-306)

4. Line 179: list the three subregions.

Reply: We agree with referee and spell out all three subregions. (Lines 182-183)

5. What is this sentence based on explicitly? Regarding the magnitude of the impact, UOGD and COGD could elevate the PR level by up to 0.13 mBq/m³ and 0.029 mBq/m³, respectively.

Reply: We detailed the calculation in Methods section Lines 144-145. Specifically, we multiplied the point estimate of the coefficient with the 95 percentiles of upwind UOGD and COGD wells within 20 km. With this method, we hoped to estimate the impacts of UOGD on PR in some "worst" cases.

Dear Referee #5,

We are grateful for your approval and your previous comments in the last round of peer-review.

We got your ideas, though you did not provide specific idea to modify the manuscript. We revised the discussion section regarding health impacts.

I hope you feel more confident of our manuscript.

Regards,

Longxiang Li

Reviewer #5 (Remarks to the Author):

Thank you for the opportunity to read the revised manuscript and rebuttal letter. It is pleasing to see that the authors found the comments from all 5 reviewers beneficial in revising the manuscript and they have made appropriate modifications. I am still not convinced by the suggested health impacts, but the authors have at least toned down their statements in relation to this. I think it appropriate that this manuscript is now published so that others can then read and judge for themselves.

Reply: Thank you for your approval. I totally understand your skepticism of the health effects of UOGD through particle radioactivity. You're far from alone. In the revised version, I further tone down the statements and admit that more field measurements are needed to validate this exposure pathway. (Lines 319-321)

Reviewer comments, third round

Reviewer #1 (Remarks to the Author):

The authors were responsive, once again, and the manuscript is improved. I particularly appreciate the time-stratified models.

-Joan Casey